# A Novel Machine Learning 13-Gene Signature: Improving Risk Analysis and Survival Prediction for Clear Cell Renal Cell Carcinoma Patients

**DOI:** 10.3390/cancers14092111

**Published:** 2022-04-24

**Authors:** Patrick Terrematte, Dhiego Souto Andrade, Josivan Justino, Beatriz Stransky, Daniel Sabino A. de Araújo, Adrião D. Dória Neto

**Affiliations:** 1Bioinformatics Multidisciplinary Environment (BioME), Metropole Digital Institute (IMD), Federal University of Rio Grande do Norte (UFRN), Natal 59078-400, Brazil; dhiego.souto.072@ufrn.edu.br (D.S.A.); josivan.justino@unir.br (J.J.); beatriz.stransky@ufrn.br (B.S.); daniel@imd.ufrn.br (D.S.A.d.A.); adriao@dca.ufrn.br (A.D.D.N.); 2Department of Engineering and Technology (DETEC), Pau dos Ferros Multidisciplinary Center, Federal Rural University of Semi-arid (UFERSA), Pau dos Ferros 59900-000, Brazil; 3Department of Mathematics and Statistics (DME), Federal University of Rondônia (UNIR), Ji-Paraná 76900-726, Brazil; 4Biomedical Engineering Department, Center of Technology, UFRN, Natal 59078-970, Brazil; 5Department of Computer Engineering and Automation, UFRN, Natal 59078-970, Brazil

**Keywords:** kidney cancer, clear cell renal cell carcinoma (ccRCC), gene signature, prognosis, survival analysis, feature selection, mutual information, machine learning

## Abstract

**Simple Summary:**

Clear cell renal cell carcinoma is a type of kidney cancer which comprises the majority of all renal cell carcinomas. Many efforts have been made to identify biomarkers which could help healthcare professionals better treat this kind of cancer. With extensive public data available, we conducted a machine learning study to determine a gene signature that could indicate patient survival with high accuracy. Through the min-Redundancy and Max-Relevance algorithm we generated a signature of 13 genes highly correlated with patient outcomes. These findings reveal potential strategies for personalized medicine in the clinical practice.

**Abstract:**

Patients with clear cell renal cell carcinoma (ccRCC) have poor survival outcomes, especially if it has metastasized. It is of paramount importance to identify biomarkers in genomic data that could help predict the aggressiveness of ccRCC and its resistance to drugs. Thus, we conducted a study with the aims of evaluating gene signatures and proposing a novel one with higher predictive power and generalization in comparison to the former signatures. Using ccRCC cohorts of the Cancer Genome Atlas (TCGA-KIRC) and International Cancer Genome Consortium (ICGC-RECA), we evaluated linear survival models of Cox regression with 14 signatures and six methods of feature selection, and performed functional analysis and differential gene expression approaches. In this study, we established a 13-gene signature (AR, AL353637.1, DPP6, FOXJ1, GNB3, HHLA2, IL4, LIMCH1, LINC01732, OTX1, SAA1, SEMA3G, ZIC2) whose expression levels are able to predict distinct outcomes of patients with ccRCC. Moreover, we performed a comparison between our signature and others from the literature. The best-performing gene signature was achieved using the ensemble method Min-Redundancy and Max-Relevance (mRMR). This signature comprises unique features in comparison to the others, such as generalization through different cohorts and being functionally enriched in significant pathways: Urothelial Carcinoma, Chronic Kidney disease, and Transitional cell carcinoma, Nephrolithiasis. From the 13 genes in our signature, eight are known to be correlated with ccRCC patient survival and four are immune-related. Our model showed a performance of 0.82 using the Receiver Operator Characteristic (ROC) Area Under Curve (AUC) metric and it generalized well between the cohorts. Our findings revealed two clusters of genes with high expression (SAA1, OTX1, ZIC2, LINC01732, GNB3 and IL4) and low expression (AL353637.1, AR, HHLA2, LIMCH1, SEMA3G, DPP6, and FOXJ1) which are both correlated with poor prognosis. This signature can potentially be used in clinical practice to support patient treatment care and follow-up.

## 1. Introduction

Renal cell carcinoma (RCC) occurs in the renal cortex or the renal tubular epithelial cell. The molecular subtypes of renal cancers are clear cell RCC (ccRCC), papillary RCC (pRCC), and chromophobe RCC (ChRCC). RCC accounts for more than 90% of cancers in the kidney [1], of which 80–90% are ccRCC [2], and more than 30% of patients with ccRCC experience metastasis [3]. In 2020, the worldwide mortality rate from kidney cancer was an estimated 179,368 cases International Agency for Research on Cancer (IARC). The American Cancer Society estimated a prevalence of 76,080 new cases of kidney cancer for 2021 in the United States (48,780 in men and 27,300 in women), and an estimated mortality rate of 13,780 people (8790 men and 4990 women) [4]. Depending on the stage at diagnosis, the five-year survival rates of RCC in the US are the following: 93% for localized disease (stage I), 72.5% for regional disease (stage II/III, local lymph node involvement), and only 12% for late-stage (stage IV metastatic) [5]. The poor survival outcomes of metastatic patients with ccRCC reveal the importance of seeking new and robust biomarkers of prognosis, and of preventing the progression of non-metastatic tumors.

The challenges of artificial intelligence (AI) applications to cancer care are driven by the translation of models with clinical validity, utility, and usability into feasible clinical treatment [6]. In the field of precision medicine applied to cancer, feature selection is useful in detecting the most important traits and molecular profiles for predicting the survival risks of a patient′s outcome through a given gene set. A gene signature is a set of genes whose expression pattern in a specific cell type and condition can provide a biomarker for diagnosis, prognosis, or therapeutic responses in cancer patients [7]. The gene signatures can be defined by the pattern of the Single Nucleotide Variant (SNV) mutational profile; the copy number of alterations (CNA); the methylation levels; or the expression of messenger or other RNA types. Genes involved in the biological processes of many tumors might be overexpressed or inhibited, signaling a better or worse prognosis for the patient [8]. While most of the studies used only mRNA data to build their signatures, microRNA and/or clinical data can be explored as relevant features to build a predictive signature [9,10,11,12,13,14].

Nowadays, the scientific community is still searching for new biomarkers for ccRCC, and feature selection methods using survival analysis provide a robust exploratory methodology before experimental validations. Survival analysis is a field of statistics that predicts the time until an event of interest happens in many domains [15]. The most commonly used method for survival analysis is the Cox Regression model [16]. The Cox model is semi-parametric, that is, the distribution of the event of interest is unknown. In addition, Cox models are widely used for censored data, i.e., when the event is not observed during the study period due to loss to follow-up, study termination, or the patient’s death by other causes. Regularized Cox models provide suitable predictions for high-dimensional data using penalty functions with the main regularizers Lasso-Cox, Ridge-Cox, and Elastic net-Cox [15]. Ensemble learning methods are committees of machine learning models, in other words, they combine the majority of the votes for each model in an ensemble or they adjust the weighted vote of each model. Moreover, this approach results in a more robust, efficient, and stable model compared to singular models. In this work, we applied Cox models and ensemble methods using gene expression to predict the overall survival (OS) after diagnosis of ccRCC.

Lasso-Cox regression generated most of the reviewed gene signatures for ccRCC [9,10,13,17,18,19]. All the studies reviewed in this work use the TCGA-KIRC dataset to train and validate the results. Fewer studies validated their results with other datasets such as GEO database [2,10,13], ICGC-RECA [2,11], and data from Fudan University Shanghai Cancer Center (FUSCC). The most common methodologies used to discover and validate gene signatures were differentially expression analysis (DEA), and gene set enrichment analysis (GSEA). Only one study compared its methodology to three other biomarker signatures from our literature selection [9]. In addition, there was a lack of comparisons between the gene signatures. As far we know, our study presents the most comprehensive comparison between gene signatures, including ensemble methods, machine learning, and feature selection. 

This study aims to specify a gene signature based on the state-of-the-art algorithms of feature selection methods, and to be able to predict the survival risk of ccRCC patients. Moreover, this study compares the novel signatures obtained by these feature selection methods, and other previously published gene signatures. The best-performing gene signature was achieved using the mutual-information-based ensemble method of min-Redundancy and Max-Relevance (mRMR) [20]. Specifically, the mRMR is an ensemble-based method to select a minimal set of features with a maximum prediction performance. The flowchart shown in Figure 1 displays a summarized view of the discovery process for the novel mRMR gene signature of ccRCC.

## 2. Materials and Methods

### 2.1. Literature Search Using PubMed

A literature search for gene signatures was conducted using PubMed to select studies from 2015 to 2020 (Figure A1), given that the search was carried out in January 2021, from this date, we performed the analyses and wrote the manuscript. The majority of papers were published in the last five years since 2020, therefore we excluded the period of 2008 to 2014. The PubMed query of terms comprised the following: (renal OR kidney) AND (clear cell) AND (cancer) AND (prognosis OR survival OR outcomes) AND (regression) AND (gene signature).

The search query resulted in 77 papers, and we adopted the following as inclusion criteria: original articles on human ccRCC about survival prognosis or tumor staging classification. The exclusion criteria consisted of the following: reviews, editorials, conferences, or abstracts; studies about other RCC subtypes, such as pRCC, ChRCC, or Sarcomatoid renal cell carcinoma; and studies that evaluated genes based on their corresponding patient prognoses depending on chosen treatment, on biomarkers predicting treatment resistance, or on tolerance to renal allograft. Ultimately, we adopted 14 gene signatures with a total of 221 unique genes (Table A1).

### 2.2. Data

From a bottom-up perspective, this work is data-driven by the gene expression and survival data of the larger public dataset of ccRCC (*n* = 530), The Cancer Genome Atlas Consortium of Kidney Renal Clear Cell Carcinoma (TCGA-KIRC) [21,22]. For external data validation, in order to corroborate the findings within our novel gene signature, we used the dataset of ccRCC samples (*n* = 91) from the International Cancer Genome Consortium (ICGC-RECA) [23,24].

### 2.3. Data Pre-Processing

Data pre-processing was undertaken to select the genes from both TCGA-KIRC (*n* = 60,489) and ICGC-RECA (*n* = 49,221) cohorts to obtain a consensus nomenclature for the genes in the signatures, and to map the latter with the HGNC Symbol and the Ensembl identifiers. The reference genomes used for the TCGA-KIRC and ICGC-RECA databases are the GRCh38 and the GRCh37 genomes, respectively. Despite this distinction, both reference genome versions are highly concordant [25].

For both datasets, we used unprocessed raw count data, and to reduce the batch-effect of datasets, we evaluated the following normalization methods: (1) scaling to the range interval between zero and one; (2) variance stabilizing transformation with DESeq2 [26]; and (3) Box-Cox normalization with Caret R package [27] (v. 6.0–90). The chosen method was Box-Cox transformation, with the higher correlation R = 0.97 between the median of each gene expression of datasets (Figure A2).

### 2.4. Feature Selection with Bioinformatics Analyses and Machine Learning

From a top-down perspective, in order to guide our feature selection, we performed two differential expression analyses of RNA-Seq with DESeq2 [26]. The first analysis was to compare solid normal tissue (NT) samples (*n* = 71) versus primary solid tumor (TP) samples (*n* = 530) using the absolute log2 fold-change (LFC) >3 and *p*-value adjusted (FDR) <0.01, from which we obtained 1775 genes that were under- and over-expressed. The LFC of each gene expression is the ratio of the mean normalized by log2 in the two groups of samples. The second analysis was to compare the non-metastatic (M0) samples (*n* = 422) against metastatic (M1) samples (*n* = 78); then using absolute LFC over 2 and *p*-value adjusted (FDR) <0.01, we obtained 156 altered genes.

To optimize the right candidates to their ideal biomarker genes, we also included 221 genes from the literature, and selected 1259 tissue-specific genes of Kidney Cortex tissue with significant expression quantitative trait locus (eQTL) obtained from the Genotype-Tissue Expression (GTEx) Project [28,29]. The feature selection methods and supervised Cox regression models were then trained by gene expressions of 3284 pre-selected genes, the overall survival (OS) days since the diagnosis, and the OS status (deceased or living) of TCGA-KIRC patients.

Inspired by the methodology of [30], we produced the new gene signatures using 6 feature selection methods divided into two main categories:Filtering methods of feature importance: Extreme Gradient Boosting (XGBoost), Generalized Boosted Regression Model (GBM), and Recursive Partitioning for Survival Trees (Rpart).Wrapper methods: Minimum Redundancy Maximum Relevance (mRMR); Recursive Feature Elimination (RFE); and Boruta.

For the filtering methods, we selected the 30 most important genes for patient survival. We chose the number of 30 genes based on this being the average number of genes in signatures referenced in the literature. The wrapper methods selected the most important genes based on the best performing metrics of models without predefining the number of genes on signatures. The new gene signatures generated by each feature selection are presented in Table A2.

We evaluated the signatures using Machine Learning analyses of eight linear survival models with optimized auto-tuning hyper-parameters: Extreme Gradient Boosting (XGBoost), Cox Model with gradient boosting (GLMBoost), Generalized Boosted Regression Model (GBM), Cox Proportional Hazards Regression Model (CoxPH), Gradient Boosting with Regression Trees (Blackboost), and the three models Penalized Cox Regression (glmnet) [31]–LASSO, ElasticNet and Ridge regression. Each model calculates the fitted coefficient for each gene.

The mRMR method applies mutual information to select features that maximize the statistical dependency on the joint distribution of the target variable of supervised learning [20,32]. The maximum relevance for the feature set *S*, given the mutual information of gene *g_i_* in *k*-classes, is:
(1)
$$maxD\left(S,k\right),D=\frac{1}{\left|S\right|}\underset{g_{i}\in S}{\sum }I\left(g_{i},k\right)$$


The minimum redundancy in the feature subset condition is given by the sample vectors of all genes *g_i_*, *g_j_*:
(2)
$$minR\left(S,k\right),D=\frac{1}{\left|S^{2}\right|}\underset{g_{i},g_{j}\in S}{\sum }I\left(g_{i},k\right)$$


This work uses the implementation of the R package mRMRe [33] (v. 2.1.2) available in CRAN on expression data. The target features consisted of the overall survival days and overall survival status. We set an ensemble of 5 executions filtering 20 genes per run, resulting in a set of 64 unique genes as relevant features. Finally, we performed a forward search feature selection with variable ranking based on mutual information difference of the most representative genes with respect to AJCC Staging, resulting in a 13-gene signature (Figure A3).

The framework of Tidyverse in R (v. 4.1.1) was used for pre-processing, and the framework mlr3 (Machine Learning in R) [34] carried out the evaluation of the metrics of feature selection and model benchmark. All of the code for the experiments was written in R. For the multicollinearity analysis, we built the visualization with corrplot [35] (v. 0.92), and we assessed the degree of collinearity among independent variables. None of the genes had Variance Inflation Factors > 5 (Figure A4). Additionally, no correlations greater than or equal to 0.7 were found between the genes (Figure A5). For the Variable Ranking Based on Mutual Information Difference, we used the R package varrank [36] (v. 0.4).

### 2.5. Model Evaluation and Statistical Analysis

The concordance C-index is a commonly used metric, but is not a proper strategy to predict the t-year risk of an event [37]. Therefore, to evaluate the performance of each survival model, we applied the measure of the area under the time-dependent ROC curve (AUC Uno) [38]. For internal validation, we used AUC Uno of 10 years on 3-fold cross-validation of TCGA-KIRC in 100 repetitions. For external validation, we used AUC Uno of 7-years by training with TCGA-KIRC and predicting the ICGC-RECA dataset using 100 repetitions through censored regression models. We restrict the 10-year prediction for TCGA-KIRC to exclude outliers in the long tail of the density plot of the patient’s overall survival. For the ICGC-RECA dataset, we decided to maintain a 7-year prediction in order to include all samples, and limit the time prediction to the range of distribution of this dataset for external validation (Figure A6). The sensitivity (SE) and the specificity (SP) describe the distinguishing risk of patients to be deceased by time *t* from those who will be alive, with values ranging from 0 to 1, where 1 corresponds to the best model performance, and 0.5 represents a random prediction. The evaluation was performed with the R package survAUC [39] (v. 1.0–5).

The Kaplan–Meier analysis is the main visualization graph used to distinguish between high-risk, moderate, and low-risk patients. The *p*-value was calculated by the log-rank test using the survminer [40] (v. 0.4.9) R package and by comparing the predicted survival distributions of groups′ high, moderate, and low risk.

The enrichment analysis was performed using the 13-gene signature on the curated database of DisGeNET [41] (v7.0) with gene-disease associations (GDAs) filtering by FDR (<0.05).

The flowchart was created using diagrams.net. The figures were implemented in R 4.1.1 using the following packages: VennDiagram [42] (v. 1.7.1); the ggplot2 (v. 3.3.5) for Volcano plots, Heatmap and Boxplots; GOplot [43] (v. 1.0.2) for the circular visualization of mRMR genes and sets of genes; FactoMineR [44] (v. 2.4) and factoextra [45] (v. 1.0.7) for the principal component analysis (PCA); survival [46] (v. 3.2–11) and ggstatsplot [43] (v. 0.9.0) for the Aalen′s additive cox regression; clusterProfiler [47] (v. 4.2.1) and disgenet2r [41] (v. 0.99) for the enrichment analysis with a Heatmap-like functional classification; survminer [40] (v. 0.4.9) and finalfit [48] (v. 1.0.4) for the survival curves and the Forest plot for Cox proportional hazards model; and pheatmap [49] (v. 1.0.12) for the Heatmap with Hierarchical clustering of RNA-seq expression and clinical annotation with dendrograms.

## 3. Results

### 3.1. Clinical Characteristics of the ccRCC Cohorts

To produce our gene signature, we used the TCGA-KIRC (*n* = 530) and ICGC-RECA (*n* = 91) samples of RNASeq data of ccRCC. The characteristics of both cohorts for training and validation datasets are summarized in Table 1. The clinical characteristics with their respective *p*-value tests indicate that there is no significant distinction in the distributions between both datasets, except for Neoplasm.

### 3.2. mMRM Gene Selection

The mRMR executed a supervised gene selection of 3304 genes with four clinical features: overall survival (OS) days, OS status, age and sex. To identify the most representative genes of the signature related to Stage AJCC, we performed a forward search feature selection Variable Ranking Based on Mutual Information Difference, resulting in a 13-gene signature (AR, AL353637.1, DPP6, FOXJ1, GNB3, HHLA2, IL4, LIMCH1, LINC01732, OTX1, SAA1, SEMA3G, ZIC2–Figure A3) able to predict distinct outcomes (high, moderate, and low survival risk) of patients with ccRCC. To select the best independent predictors genes for survival risk, it is important to avoid multicollinearity; therefore, we assessed the degree of collinearity among independent variables. None of the genes had Variance Inflation Factors > 5 (Figure A4). Additionally, no correlations greater than 0.70 were found between the genes (Figure A5).

We visualized the composition of filtered genes with a Venn diagram (Figure 2a) with the intersection sizes of genes and the original sets of genes. In particular, most of the mRMR genes (*n* = 7) were obtained from the differential gene expression analysis (DEA) comparing normal tissues versus primary tumor samples (Figure 2b), with a larger number of upregulated genes, including the mRMR genes HHLA2, LINC01732, SAA1, AL353637.1, and ZIC2. The downregulated mRMR genes for normal versus tumor samples are DPP6 and FOXJ1. The DEA of comparing non-metastatic versus metastatic samples (Figure 2c) identified less differentiated genes (*n* = 2), with the upregulated genes OTX1 and ZIC2. The genes selected with mRMR on TCGA-KIRC samples are presented in Figure A7 with a circular visualization of the relationship between genes and their original sets of DEA, genes from GTEx portal of expression quantitative trait loci (eQTLs) in Kidney Cortex, and gene signatures from the literature.

### 3.3. Performance of the Feature Selection Models for Internal and External Validations

To compare our mRMR signature with six feature selection methods (Recursive Feature Elimination, Boruta, Rpart, GBM and XGBoost for Survival) and 14 signatures published, we performed a benchmark using eight survival models of cox survival regressions (XGBoost, GLMBoost, Gbm, CoxPH, Blackboost, Ridge, Elastic Net, and Lasso). The benchmark results are shown in Figure 3a with the performance of 100 repetitions of predictions with Area Under the Curve (AUC) Receiving Operator Characteristics (ROC) Uno evaluating the 20 gene signatures using 3-fold cross-validation of TCGA-KIRC dataset. We can observe that the model Lasso-Cox regression of glmnet had the best mean AUC, 0.81, in internal validation for mRMR. The minimal set of genes with best performance to predict TCGA-KIRC as internal validation is: AL353637.1, DPP6, FOXJ1, GNB3, HHLA2, IL4, LIMCH1, OTX1, SAA1, and ZIC2.

Figure 3b shows the boxplot of the results of external validation in 100 random repeats. The upper plot also displays the mean of the adjusted *p*-value of the log-rank test of survival risk. Please note that the only signature that has a significant adjusted *p*-value (*p* < 0.05) is the mRMR. The lower plot displays the AUC metric of each survival prediction, and the number displayed on boxplots is the average value of all repeats. Please note that the best mean of AUC is 0.71 for the mRMR signature. The minimal set of genes for training with samples of TCGA-KIRC and predicting the survival risk of samples of ICGC-RECA is AR, AL353637.1, FOXJ1, HHLA2, SEMA3G, and LINC01732.

In Figure 4, we display the Kaplan–Meier curves and a principal component analysis (PCA) of two random predictions of internal and external validations.

For internal validation of the mRMR gene signature, we performed 3-fold cross-validation with AUC assessed on TCGA-KIRC with time-dependent intervals of seven years. Figure 4a shows a prediction of a random 33% sampling from TCGA-KIRC after training the regression model with 66% of the samples. The Kaplan–Meier curves (Figure 4a) are evaluated by the *p*-values of the log-rank test, indicating the separation between patients with high, moderate, and low risk. Figure 4a displays a PCA with the same predicted samples using only the expression of mRMR genes. Please note that only one patient was deceased in the low-risk group, and there is a visible separation between the low-risk and high-risk groups of patients on the *x*-axis of the PCA.

For external validation, we trained the model with the TCGA-KIRC dataset and predicted all the samples of ICGC-RECA. Analogously, in Figure 4c, a model trained with TCGA-KIRC data predicts ICGC-RECA samples in separated survival curves risks (*p* < 0.05) and AUC of 0.66. In Figure 4d, we performed a PCA with mRMR gene on the same previously predicted samples of ICGC-RECA, and the *x*-axis also separates the centroids of the risk clusters.

### 3.4. Biological Interpretation: Gene Contributions for Survival Risk and Enrichment Analysis

To shed light on the ability of each gene to predict ccRCC risk, we performed an additive regression, plotting the genes’ coefficients with time-varying and covariate effects. Similar to the forest plot of hazard ratio regression (Figure A8), Figure 5 shows the estimated coefficients of the increasing curves for the following significant high expression genes with a high risk of death: FOXJ1, OTX1, and IL4. On the other hand, the decreasing curves indicate that the high expression of the following genes is related to the low risk of death: AL353637.1, DPP6, HHLA2, and LIMCH1. A common classical representation of these covariate effects is the Hazard ratio in Figure 5 of the forest plot for the Cox proportional hazards model.

We confirmed the genes’ contributions to survival risk by checking protein expression according to cancer stage using the UALCAN dataset of ccRCC from Clinical Proteomic Tumor Analysis Consortium (CPTAC). As a result, the gene expression by overall survival is corroborated with the levels of protein expression and the cancer stage. In CPTAC-ccRCC, the protein expression of genes AR, GNB3, HHLA2, LIMCH1, and SAA1 had statistical significance in some comparisons of normal samples and cancer stage (Figure A9a–e) [50]. In particular, HHLA2 protein expression in samples of Stage 1 was higher than in stage 4, but normal tissue had a lower protein expression than any tumor stage (Figure A9c). This protein expression shift is compatible with our results in Figure 5 of the decreasing curve for HHLA2, and is in accordance with the TCGA-KIRC RNASeq data, since the higher expression of HHLA2 demonstrates a better prognosis (Figure A9c).

We verified patient survival curves by comparing the low/medium versus high expression of TCGA-KIRC data on UALCAN portal [51]. The above results correspond with the OS patients with low/medium versus high expression, available on the effect of expression level of patient survival. We noticed that patients with a poor prognosis had low expression of AR, DPP6, HHLA2, LIMCH1 and SEMA3G. Additionally, poor prognoses of patients can be identified with high expressions of FOXJ1, GNB3, OTX1, SAA1, and ZIC2.

Furthermore, in accordance with the above results, performing a Heatmap with hierarchical clustering combining RNA-Seq of patients from TCGA-KIRC and ICGC-RECA (Figure A10), we verified that the cluster of genes SAA1, OTX1, ZIC2, LINC01732, GNB3, and IL4, with high expression, is correlated with Stage T3 AJCC, metastasis, and poor prognoses. Likewise, the cluster of genes AL353637.1, AR, HHLA2, LIMCH1, SEMA3G, DPP6, and FOXJ1, with low expression, is correlated with poor prognoses.

To clarify the relationship between the genes and other kidney pathologies, we checked the statistical significance of multiple diseases associated with the enriched genes in the signature. Figure 6 shows a subset of 11 genes from within the signature, and most genes are related to Neoplasms, except for AL353637.1, LINC01732, and DPP6. Nevertheless, genes DPP6 and AR are enriched to clear-cell metastatic RCC diseases. We identified six genes enriched to kidney diseases and ccRCC (AR, DPP6, GNB3, IL4, SAA1, SEMA3G). Other enriched genes we found (AR, GNB3, HHLA2 and IL4) were related to transitional cell carcinoma of the bladder (also known as Urothelial carcinoma). GNB3 and IL4 are both enriched in kidney diseases, transitional cell carcinoma, and neoplasm metastasis. This enrichment analysis also confirms the results of benchmark and comparisons to the literature, indicating the importance of the selected mRMR genes in predicting ccRCC survival risk.

## 4. Discussion

From our 13-gene signature, a subset of eight genes had been reported previously in distinct signatures for ccRCC, including other recent signatures that were not compared in our benchmark: AR [19,51], SEMA3G [19,52,53], LIMCH1 [9], DPP6 [54,55], FOXJ1 [56,57], ZIC2 [11], IL4 [19,58,59,60], and OTX1 [12]. The concordance of this work with published signatures strengthens the validity of our methodology to obtain a ccRCC survival signature.

FOXJ1, IL4, HHLA2, and SEMA3G are immune-related genes [19,52,53], corroborating the high immunogenicity of ccRCC. Forkhead Box J1 (FOXJ1) is a transcription factor, and a member of the FOX family, involved in ciliogenesis. Its defective expression is associated with some inflammatory [61] and autoimmune [62,63] diseases. FOXJ1 has previously been identified as a prognostic marker of RCC, where its expression was reported to be upregulated [57]. Moreover, it has been reported to be upregulated in bladder cancer [64], hepatocellular carcinoma [65] and colorectal cancer [66]. Conversely, its low expression has been reported to be correlated with gastric cancer [67], ependymoma and choroid plexus tumors [68]. AL353637.1 is a pseudogene nearby the gene FOXB2, also belongs to the FOX family of FOXJ1 [56], and contains a variant (rs115747230) associated with chronic kidney disease [69]. Interleukin 4 (IL4) is a cytokine that induces differentiation of T cells and is present in the tumor environment of many cancers. The expression of IL4 in the tumor microenvironment can improve tumor growth and the blockade of IL4 can delay the growth [70] and can also improve immunotherapies (in mice models) such as CpG ODN or anti-OX40 AB [71]. Polymorphisms of the IL4 gene have been associated with many cancers [72]. HERV-H LTR-Associating 2 (HHLA2, also known as B7-H7) is a member of the B7-family of immune checkpoint molecules, known to perform an inhibitory activity in human CD4+ and CD8+ T cells by binding to their receptors [73,74]. It is known to have limited expression in normal tissues and to be highly expressed in cervical adenocarcinoma [75], pancreatic and ampullary cancers [76], also widely expressed in different subtypes of human lung cancer [73,77]. The 5-year survival rate of patients with gastric cancer was significantly higher in patients with HHLA2 highly expressed [78]. In particular, the overexpression of HHLA2 in patients after surgery was identified to promote ccRCC progression when compared to normal adjacent tissue [79], which corresponds with our results regarding HHLA2 expression. The knockdown of HHLA2 decreased the expression of genes related to the cell cycle, as well as the ability of the cells to migrate and invade [79]. SEMA3G belongs to the family of class-3 semaphorins, and studies indicate that this gene is linked to kidney diseases [80,81], suggesting important roles with neuropilin and plexin families in the etiology of cancer [82], and it is also an inhibitor of glioma progression by competing with VEGF for receptor NRP1 [83]. In single-cell RNA-seq study of kidney with transplant biopsy, SEMA3G activates an angiogenic program [84]. Patients with high expression of SEMA3G and AR have better prognoses according to the survival analysis of UALCAN RNASeq data [51]. The presence of immune-related genes in our signature strengthens the approach of focusing on the genes from the immune system to build a prognostic signature [19,85]. Our findings reinforce that HHLA2 is an important immune-related biomarker of ccRCC.

The genes AR, OTX1, and ZIC2 are transcription factors. In particular, Androgen Receptor (AR) is a transcription factor whose activity is highly critical to prostate cancer evolution [86]. The expression of AR-V7, its isoform, which is encoded by splice variant 7 in circulating tumor cells of prostate cancer, was reported to be associated with drug resistance [87]. AR interacts with VHL to modulate the metastasis of ccRCC [88], and AR inhibition can attenuate RCC progression [89]. The epigenetic control of AR co-regulates lysine-specific histone demethylase 1 (LSD1) in kidney cancer development, and the LSD1 inhibitor can reduce growth of kidney cancer cells [90]. Additionally, AR could suppress ccRCC cell progression by increasing the expression of circRNA circHIAT1 [91]. In addition, in vitro research and in vivo mouse model studies indicate that AR mediates lncRNA-TANAR signals that might play a crucial role in ccRCC progression and metastasis [92]. The studies above indicate that AR might be a promising drug target for treatment of ccRCC. OTX1 is a protein-coding gene of the bicoid sub-family of homeodomain-containing transcription factors, involved in differentiation of young neurons of the deeper cortical layers, and in proliferative zones of the neocortex [93]. OTX1 is related to breast cancer, medulloblastomas, colorectal cancer, hepatocellular carcinoma and bladder cancer [12]. The zinc finger of the cerebellum 2 (ZIC2) is a transcription factor with an important role in neural development and mutations of ZIC2, which could lead to brain malformations [94,95]. ZIC2 is an oncogenic with overexpression correlated with progression of epithelial ovarian tumors [96]. In breast cancer, low expression of ZIC2 has been correlated with poor outcomes and acts as a tumor suppressor by regulating STAT3 [97]. ZIC2 also upregulates gene RUNX2 and promotes ccRCC progression through inhibition of tumor suppressor NOLC1 [98].

Lim and Calponin Homology Domains 1 (LIMCH1) is an actin-stress-fiber-associated protein, a gene encoding zinc-binding protein, and is known to negatively regulate cell-spreading and migration [99]. It has been reported to be downregulated in malignant lung tissue [100] and upregulated in breast cancer [101]. LIMCH1 is upregulated with a strong association to poor prognoses, representing a potential biomarker for cervical cancer treatment [102]. According to survival analysis of the Human Protein Atlas [103], LIMCH1 is also a prognostic gene, whose high expression is associated with favorable outcomes in renal cancer [104].

Dipeptidyl Peptidase Like 6 (DPP6) is a type II membrane glycoprotein known to regulate potassium channels and is mainly expressed in the central nervous system [105]. The methylation of CG sites in the DPP6 promoter was reported to be in greater numbers in tumor samples compared to normal samples from pancreatic ductal carcinoma; thus, the hypermethylation of DPP6 promoter is associated with poor overall survival [106]. The hypermethylation of DPP6 was associated with high-grade tumor in ccRCC [55]. Additionally, high expression of DPP6 was reported to be correlated with good prognoses in patients with breast cancer [107].

Guanine Nucleotide Binding Protein Beta Polypeptide 3 (GNB3) is involved in various transmembrane signaling systems such as in GTPase activity. Some studies associate the polymorphism GNB3-C825T with cholangiocarcinoma [108] and thyroid carcinoma [109], but another study discarded a relationship with the risk for breast cancer [110].

Serum Amyloid A 1 (SAA1) is an acute-phase protein mainly produced by hepatocytes in response to infection, tissue injury and malignancy. SAA1 modulates neutrophil function in the context of cancer [111]. SAA1 gene expression in patients with RCC is associated with poor prognosis [112]. According to survival analysis of Human Protein Atlas [103], SAA1 is also a prognostic gene with high expression for unfavorable outcomes in renal cancer [113]. Moreover, multiple mutation variants of SAA1 have been identified in patients with RCC [114].

LINC01732 is affiliated with the long non-coding RNAs (lncRNAs) class. To the best of our knowledge, there are no publications regarding LINC01732 at this time. Nevertheless, increasing evidence suggests that lncRNAs play critical roles in tumor development of RCC [115]. Further research could be executed to understand other lncRNAs, including LINC01732.

Since alterations in expression of different genes from the same pathway have higher impacts on gene function, we performed an enrichment analysis and identified the pathways of urothelial carcinoma, chronic kidney disease, and transitional cell carcinoma, nephrolithiasis. Although the concurrence of RCC and urothelial carcinoma is clinically rare [116], previous studies reported the identification of clear cell tumors in general bladder carcinomas [117,118]. On nephrolithiasis, studies have whoen that kidney stones are associated with increased papillary RCC risk but not clear-cell RCC risk [119].

We compared our signature in a benchmark with fourteen other signatures already published in the literature. All of the gene signatures compared in this work use TCGA as their main training set to build their models. The studies reviewed have AUC-ROC between 0.568 to 0.884 with possible values ranging from 0 to 1, and the number of genes in each signature range from 3 to 66. Some studies use a different number of patients due to the distinct filtering approaches that the authors adopted, in addition to the updates of versions of TCGA-KIRC clinical data. The least absolute shrinkage and selection operator (Lasso-Cox) was the most-used model approach to build the signatures [9,10,13,17,18,19,120,121], but network-based models with protein–protein interaction (PPI), aside from being an elegant approach for retrieving information from data, can also be used for this purpose [122,123].

This work consists of a pure in silico and data-driven study, and other analyses could be corroborated in the future with in vitro or in vivo experiments [124]. In future works, we will expand the machine learning approach presented in this work to find potential cancer biomarkers using multiples levels of biological data available in TCGA by analyzing and integrating data of long non-coding RNAs (lncRNAs), methylation, single-nucleotide variants (SNV), and copy number variants (CNV).

## 5. Conclusions

Our main goal was to compare distinct gene signatures from the literature and generate new gene signatures using feature selection methods. We contributed by providing a list of new genes, some of them not previously reported as biomarkers for ccRCC. The gene signature created by the mRMR method achieved a score of 0.82 with AUC, being the best performer. We identified two clusters of genes with high expression (SAA1, OTX1, ZIC2, LINC01732, GNB3 and IL4) and low expression (AL353637.1, AR, HHLA2, LIMCH1, SEMA3G, DPP6, and FOXJ1) that were correlated with poor prognosis. We validated our 13-gene signature for ccRCC and confirmed our results with the literature, and by comparing each cancer stage of ccRCC with CPTAC and the survival effects of gene expression of individual genes in TCGA. We believe that further studies on the involvement of these genes in renal carcinogenic processes could improve our understanding of cancer biology. After experimental validations, new possible applications in clinical practices can benefit from the biomarker found with machine learning and feature selection.

## Figures and Tables

**Figure 1 cancers-14-02111-f001:**
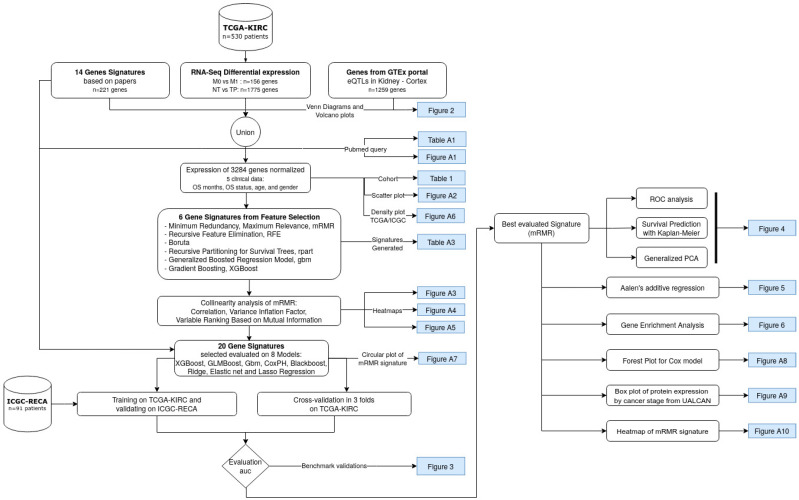
Flowchart of the current study to obtain a gene signature based on mutual information, Minimum Redundancy Maximum Relevance (mRMR). The datasets are indicated by the cylinder, white rectangles represent a step of the analysis, and the blue rectangles indicate the resulting figures and tables. TCGA-KIRC and ICGC-RECA are datasets of ccRCC.

**Figure 2 cancers-14-02111-f002:**
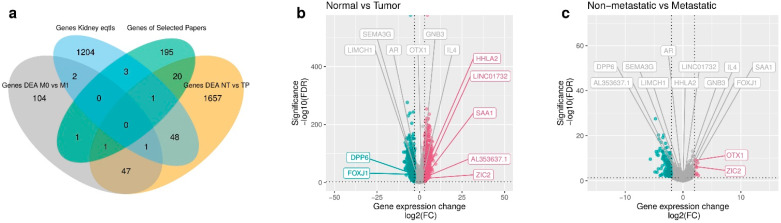
Selected genes through mRMR. (**a**) Venn diagram of prefiltered gene sets. A total of 3284 prefiltered genes is given by the sets of DEA between non-metastatic versus metastatic (156), normal tissues versus primary tumor (1775), genes from literature (221), significant eQTLs genes (1259), and 124 genes overlapping in two or three intersections of sets. (**b**) Volcano plot of DEA comparing normal tissues versus primary tumor samples of TCGA-KIRC. In green, we see the downregulated genes of normal tissues versus primary tumors (DPP6 and FOXJ1). In red, we see the upregulated genes (HHLA2, LINC01732, SAA1, AL353637.1, and ZIC2). In gray, we see the non significant genes with low fold change. (**c**) Volcano plot of DEA comparing non-metastatic versus metastatic samples. In red, we see the upregulated genes (OTX1 and ZIC2).

**Figure 3 cancers-14-02111-f003:**
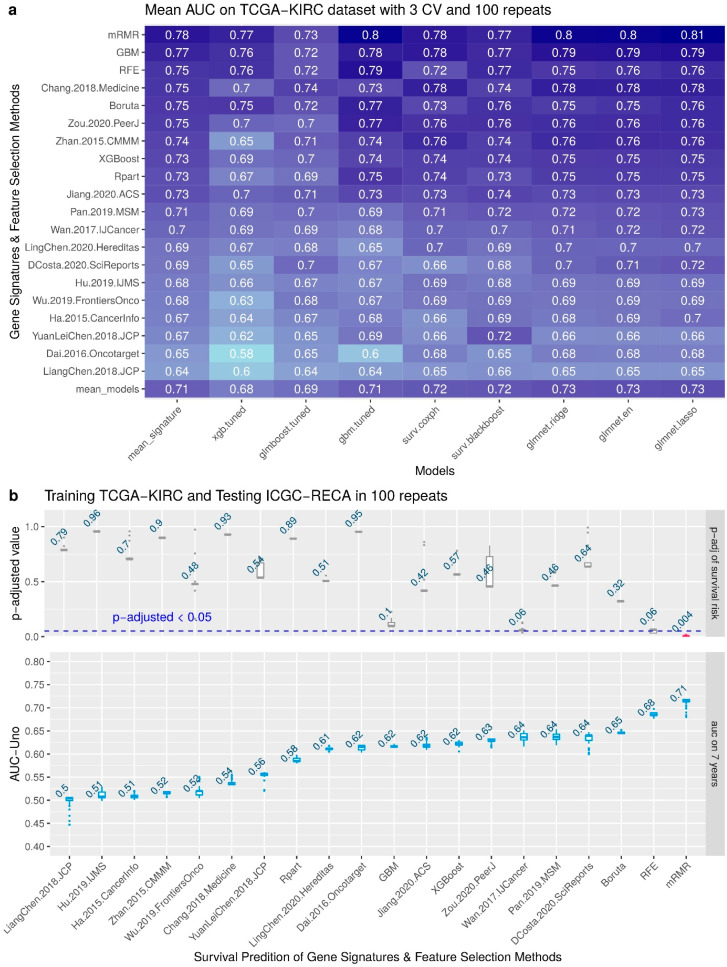
Benchmark with internal and external validation. (**a**) Comparison of 14 gene signatures from the literature and 6 feature selection on 8 models for survival risk, showing the predicted AUC of survival outcome in 10-years prediction. (**b**) Boxplots of results of each gene signature and feature selection for 7-year prediction.

**Figure 4 cancers-14-02111-f004:**
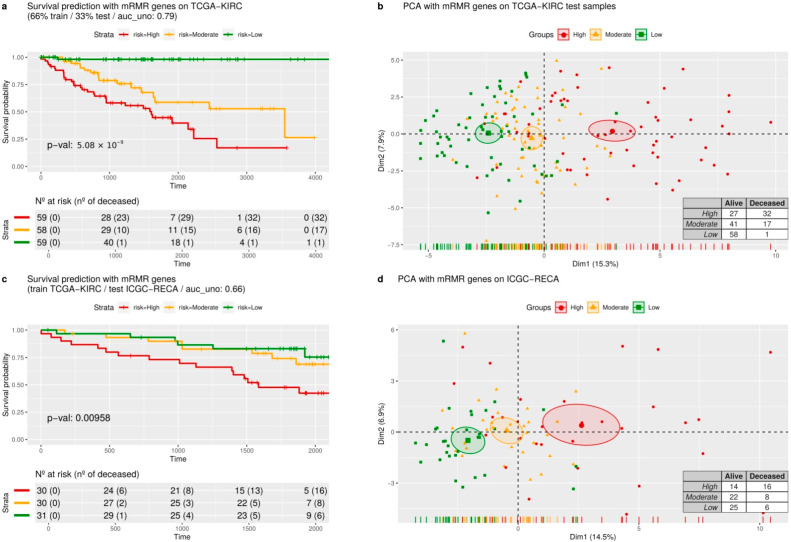
Survival risk predictions with mRMR signature and dimensionality reduction. (**a**) The survival curves are predicted in three equal-size strata of risk groups of the TCGA-KIRC dataset: higher risk (red), lower risk (green), and moderate risk (orange). (**b**) A dimension reduction of genes from the mRMR signature, using principal components analysis. (**c**) The survival curves were predicted by validating the ICGC-RECA dataset. (**d**) The principal components analysis of the ICGC-RECA dataset with genes of mRMR signature.

**Figure 5 cancers-14-02111-f005:**
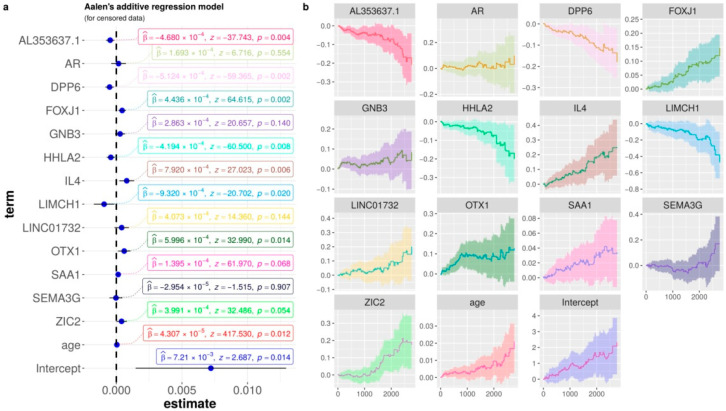
Aalen’s additive Cox regression model for censored data of the mRMR signature, and the clinical features age and metastasis. (**a**) The dot-and-whisker plots with the estimated coefficients (β), z-score, their confidence intervals (95%), and the *p*-values. (**b**) Curves of each term for the censored data in relation to time (days).

**Figure 6 cancers-14-02111-f006:**
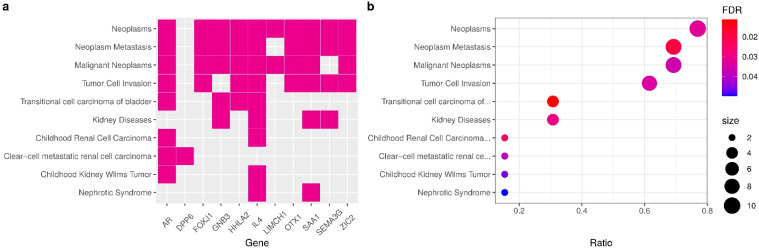
Gene enrichment analysis. (**a**) Heatmap of enriched terms and relationships of genes, displaying the fold change of differential analysis of normal tissues versus primary tumors of TCGA-KIRC samples. (**b**) Enrichment analysis of gene-disease associations (GDAs) from DisGeNET (v7.0) of expert curated databases.

**Table 1 cancers-14-02111-t001:** Study Characteristics of TCGA-KIRC and ICGC-RECA cohort with the clinical characteristics: age, gender, tumor grade, metastasis, and staging by the American Joint Committee on Cancer (AJCC).

Clinical Characteristics		Training Cohort TCGA-KIRC (*n* = 530) ^1^	Validation Cohort ICGC-RECA (*n* = 91)	*p* Value ^2^
Overall survival (days)	Mean (SD)	1343.2 (976.6)	1511.6 (634.6)	0.113
Overall survival status, *n*./total *n*. (%)	Alive	359/530 (67.7)	61/91 (67.0)	0.991
	Deceased	171/530 (32.3)	30/91 (33.0)	
Age, years	Mean (SD)	60.5 (12.0)	60.5 (10.0)	0.99
Gender, *n*./total *n*. (%)	Female	183/530 (34.5)	39/91 (42.9)	0.158
	Male	347/530 (65.5)	52/91 (57.1)	
AJCC stage, *n*./Total (%)	T1	270/530 (50.9)	54/91 (59.3)	0.343
	T2	70/530 (13.2)	13/91 (14.3)	
	T3	179/530 (33.8)	22/91 (24.2)	
	T4	11/530 (2.1)	2/91 (2.2)	
Neoplasm, *n*. (%)	N0	79 (86.8)	239 (45.1)	<0.001
	N1	2 (2.2)	16 (3.0)	
	NX	10 (11.0)	275 (51.9)	
Metastasis, *n*. (%)	M0	422/528 (79.9)	81/91 (89.0)	0.081
	M1	78/528 (14.8)	9/91 (9.9)	
	MX	28/528 (5.3)	1/91 (1.1)	

^1^ The metastasis values do not sum up to heading totals because of missing data. ^2^ The statistical tests for age and overall survival days are performed by Wilcoxon rank-sum test, and all other comparisons are by Fisher’s exact test.

## Data Availability

Publicly available datasets were analyzed in this study. The results shown here are-based upon data generated by the TCGA Research Network [125]. The TCGA-KIRC (version 07-19-2019) [126] is available via UCSC Xena Browser [22,127], and the ICGC-RECA is available via ICGC Data Portal [23,24]. Code used for analyses and to produce the figures is publicly available at: https://github.com/terrematte/gene_signature (accessed on 1 March 2022).

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
