# Peer review of "A Novel Machine Learning 13-Gene Signature: Improving Risk Analysis and Survival Prediction for Clear Cell Renal Cell Carcinoma Patients"

_cancers, 2022, doi:10.3390/cancers14092111_

Round 1

Reviewer 1 Report

The entire study was based on data mining, therefore the lack of independent validation with wet lab experiments using cell lines or clinical specimens weakens this study.
Please validate your results with functional experiments.

Reviewer 2 Report

Title: A Novel Machine Learning 13-Gene Signature: Improving Risk

Analysis and Survival Prediction for Clear Cell Renal Cell Carcinoma Patients

Comments to the Author

The work presented in the manuscript aims to predict of gene signature based on feature selection algorithms in combination with survival analysis focusing on Renal Cell Carcinoma (ccRCC). The authors used exclusively available public data Cancer Genome Atlas (TCGA-KIRC) and International Cancer Genome Consortium 27 (ICGC-RECA) for this work.  The authors have performed numerous machine learning, feature selection as well as downstream analysis for identifying the gene signature. The approach is interesting and highly relevant to current trends in biomarker research for cancer diagnosis. The manuscript is mostly well written. But I think to improve the quality of the manuscript, certain  parts need to be changed and explained clearly.

Major comments

  1. The introduction section is rather too small and lacks to explain some important knowledge in the field for the background readers. For example, why ensemble-based feature selection methods are useful, bring some literature using them and more examples of gene signatures in CCRC or related cancer types.
  2. In general Figures need to be of better quality and more explanation is needed in the figure legend. In Figure, the authors need to describe the Venn diagram and short description of what the readers should interpret. For example, the overall between all the sets are zero. What does this denote? In addition, there is no explanation about the circo plot? Can this plot be moved to the supplementary? Because Figure 2 has so many plots and seems a bit cluttered.
  3. Regarding literature search for gene signatures, why the pubmed search was limited to period of 2015 to 2020? Is there any specific reason?
  4. In the case of feature selection, Bioinformatics, and machine Learning analysis: The package information and methods were explained clearly. However, the count of preselected genes is stated as 3304. But the addition of differentially expressed gene + literature signatures+ eQTLs genes, the total number of genes don’t correspond. That is, DEG between normal % tumor is 1775, and between M0 & M1 is 422. Genes from literature is 221 and genes with significant eQTLs are 1259. This total count is 3677. Therefore, authors should clarify how the 3304 genes were selected.
  5. There is a difference in the survival time for the two datasets. For TCGA data it is 10 years and for ICGC data it is 7 years. The authors need to clarify how they accounted for the difference in the time between the datasets, especially when considering survival analysis.
  6. The authors should pay attention for grammatical errors and incomplete sentences. For instance, Clinical Characteristics of the ccRCC Cohorts: table of sample comparison shown. The wrong short form is written here, “The mRMR executed a supervised gene selection of 3304 genes with three clinical features: overall survival (OS) days, OS status, age and sex.” In this sentence, there is mention of 3 clinical features and 4 features were listed. Authors should also make the citations in the right place. For example, in page 13, in the sentence “Guanine Nucleotide Binding Protein Beta Polypeptide 3 (GNB3) is involved in 462 various transmembra[104]ne signaling systems such as in GTPase activity” the citation is in the wrong place. Similary in the discussion section, the following sentence should be re-written or removed. “As next steps, we are applying this approach of machine learning and feature selection to 500 find potential cancer biomarkers in multiples levels of biological data available in 501 TCGA, such as long non-coding RNAs (lncRNAs), methylation, single-nucleotide 502 variants (SNV), and copy number variants (CNV).”

As the article is not focusing and rather the readers do not know what are the results so far with this type of analysis. Like wise in the conclusion section “We obtained satisfactory results combining RNASeq data (ICGC and TCGA), survival information, and machine learning strategies.” This sentence is too  general, authors should write what is satisfactory?   

Reviewer 3 Report

In this study, the authors proposed a machine learning method with a 13-gene signature that could indicate clear cell renal cell carcinoma patient survival in high accuracy.

I have one main concern:

The author has done a very good job in analyzing the data, used various methods to find the signature genes, however, the authors did make it clear what the purposes are of this study, and what will the study contribute to? what are the contributions of this study to the research/clinical communities?
